# Nutritional Risk and Sarcopenia Features in Patients with Crohn’s Disease: Relation to Body Composition, Physical Performance, Nutritional Questionnaires and Biomarkers

**DOI:** 10.3390/nu15163615

**Published:** 2023-08-17

**Authors:** Konstantinos Papadimitriou, Paraskevi Detopoulou, Konstantinos Soufleris, Gavriela Voulgaridou, Despoina Tsoumana, Panagiotis Ntopromireskou, Constantinos Giaginis, Ioanna P. Chatziprodromidou, Maria Spanoudaki, Sousana K. Papadopoulou

**Affiliations:** 1Faculty of Health and Rehabilitation Sciences, Metropolitan College of Thessaloniki, University of East London, 546 24 Thessaloniki, Greece; 2Department of Clinical Nutrition, General Hospital Korgialenio Benakio, 115 26 Athens, Greece; viviandeto@gmail.com; 3Gastroenterology-Oncology Department, Theageneio Anticancer Hospital of Thessaloniki, 546 39 Thessaloniki, Greece; ksoufleris@yahoo.gr; 4Department of Nutritional Sciences and Dietetics, School of Health Sciences, International Hellenic University, 570 01 Thessaloniki, Greece; gabivoulg@gmail.com (G.V.); tsoumanadespoina@gmail.com (D.T.); mariaspan@nutr.ihu.gr (M.S.); 5Department of Food Science and Nutrition, School of Environment, University of Aegean, 811 00 Myrina, Greece; ntopromireskou@gmail.com (P.N.); cgiaginis@aegean.gr (C.G.); 6Department of Public Health, Medical School, University of Patras, 265 04 Patras, Greece; ioannachatzi@med.upatras.gr

**Keywords:** inflammatory bowel disease, malnutrition, malabsorption, prognosis

## Abstract

Patients with Crohn’s disease (CD) face malnutrition risk, which, combined with inflammation, can lead to sarcopenia, associated with a worse prognosis. The purpose of the present study was to assess malnutrition and sarcopenia in patients with CD. Fifty-three patients (26 women) participated (38.1 ± 10.9 years, 79% in remission). Body composition, physical performance, nutritional questionnaires, and biomarkers were performed. Malnutrition was screened with the Mini Nutritional Assessment (MNA) and the Malnutrition Inflammation Risk Tool (MIRT) and was assessed with the Global Leadership Initiative on Malnutrition (GLIM) tool using etiologic along with three different phenotypic criteria: low Body Mass Index (BMI), low Calf Circumference (CC), and low Fat-Free Mass Index (FFMI). To find cases and evaluate sarcopenia, the Sarcopenia Questionnaire (SARC-F) and European Working Group on Sarcopenia2 (EWGSOP2) criteria were used. Malnutrition rates were 11.3% (n = 6), 7.5% (n = 4), and 5.6% (n = 3) based on low FFMI, CC, and BMI, correspondingly. Four (7%) patients had low Hand-Grip Strength (HGS), n = 8 (14.8%) had low Appendicular Lean Mass (ALM), and n = 3 (5.6%) had low gait speed. No-one had sarcopenia. A high albumin and triceps skinfold pattern, identified by principal component analysis, was related to reduced C-Reactive Protein (CRP) levels (B = −0.180, SE = 0.085, *p* = 0.045). In conclusion, based on the studied anthropometric, nutritional, and functional variables, CD patients were not diagnosed with sarcopenia in the present study. Body composition patterns were related to the inflammatory burden, underlying the interplay of inflammation and malnutrition, even in remission states. Further studies on older populations and during disease exacerbation are necessary to explore the potential link between CD, inflammation, and sarcopenia.

## 1. Introduction

Inflammatory Bowel Disease (IBD) represents a severe and complex spectrum of gastrointestinal diseases characterized by chronic or recurrent inflammation [1], impacting health status, quality of life, and health costs [2,3,4]. Crohn’s Disease (CD) and Ulcerative Colitis (UC) constitute the two primary conditions within the spectrum of IBD, differentiated by their specific anatomical locations and patterns of lesions [1]. While the exact cause of IBD remains unclear, it is widely believed to result from a combination of genetic predisposition, alterations in the gut microbiota, and environmental factors [5]. Several lines of evidence support that dietary intake, especially a westernized diet, affects the gut microbiome, immunity, and tissue damage, being implicated in the development of IBD [6,7]. 

Regarding the nutritional status of patients with CD, they are at risk of malnutrition due to reduced food intake, malabsorption, increased gastrointestinal losses, and side effects of medication [8,9]. Patients with CD have varying degrees of malnutrition due to the different severity of enteropathy and other factors [10], with reported rates ranging from 12 to 85% [11]. Several criteria have been used to adequately screen and assess malnutrition [10], with some tools being developed specifically for patients with CD, such as the Malnutrition Inflammation Risk Tool (MIRT) [12]. Moreover, there are scarce data regarding the use of the newly developed Global Leadership Initiative on Malnutrition (GLIM) criteria for malnutrition assessment in patients with IBD [13,14].

Malnutrition can result in loss of muscle mass and function and possibly sarcopenia [15], defined by changes in muscle quantity and functionality. Indeed, it is estimated that~ 50% of patients with CD have sarcopenia [16], while estimates may vary according to sarcopenia definition, age, and ethnicity from 20 to 70% [17]. As documented by Ünal et al., a considerable proportion of IBD patients (142 out of 344) in clinical remission who are malnourished or at risk of malnutrition have a high rate of sarcopenia and probable sarcopenia [18]. Similarly, Ryan et al. showed that 42% of the studied IBD patients had sarcopenia [19]. Moreover, sarcopenia, as a direct result of chronic inflammation and malnutrition, has both diagnostic and prognostic significance in IBD patients [20]; it hinders the patient’s ability for postoperative recovery, increases the likelihood of surgical complications [19], and is generally associated to adverse patient outcomes [21]. In addition, the associated muscle wasting and weakness result in fatigue and reduced quality of life, both of which are prevalent in people living with CD [4,22].

However, there are severe biases and flaws in that kind of study. A main flaw is the heterogeneity of the sarcopenia assessment because of the plethora of existing detection tools [23]. Additionally, a noteworthy factor pertains to patients with CD, who are administered anti-Tumor Necrosis Factor (anti-TNF) agents like infliximab [24]. These individuals exhibit a reversal of symptoms associated with inflammatory sarcopenia and muscle wasting [24]. Hence, it is essential to conduct concurrent screening for nutritional status and body composition analysis in patients with IBD. This approach ensures the provision of suitable nutritional support, even during the remission period, and helps prevent sarcopenia-related surgical complications and unfavorable clinical outcomes [18].

The early detection and management of both malnutrition and sarcopenia are of utmost importance since they are both connected to postoperative complications, increased infection rates, and worse prognosis [10,21]. Thus, the aim of the present study was to assess the malnutrition and sarcopenia features in a sample of patients with CD and search for interrelations with anthropometrics, physical performance measurements, nutritional questionnaires, and biomarkers.

## 2. Materials and Methods

### 2.1. Participants 

Participants (n = 53, 27 men and 26 women) were recruited from a public Greek hospital and were assessed without comparisons with a control group. Inclusion criteria were (a) confirmed CD patients in remission or mild or moderate symptoms, (b) aged ≥18 years old and (c) capable to self-serve in their daily routine activities, (d) physical activity at least two times per week, (e) controlled diet and (f) in pharmaceutical treatment [25]. Exclusion criteria were (a) CD patients in exacerbation, (b) age less than 18 years old, (c) co-presence of other autoimmune disease, and (d) difficulties in daily living. All measurements and questionnaire filling were conducted in Theageneio anti-cancer hospital from May to August of 2022.

### 2.2. Ethics

Before proceeding to the measurements, all the patients were informed about the study’s process and the safety of the study. Then, a consent form was signed to ensure patients’ participation. The protocol was carried out in accordance with the Declaration of Helsinki (1989) of the World Medical Association. The protocol of the study was approved by the Scientific Board of Theageneio Hospital with approval code: 3734/2-3-2022.

### 2.3. Measurements

Considering the extensive array of screening tools [18,23] and measurements for detecting sarcopenia, in alignment with the new trends set by the European Working Group on Sarcopenia2 in Older People (EWGSOP2) [23,26] and the absence of a confirming structured approach for determining which tool to utilize, several measurements were taken regarding anthropometric, body composition physical performance tests, malnutrition assessment, nutritional questionnaires, biomarkers, and other variables. 

#### 2.3.1. Anthropometrics

The body weight was measured on a scale without shoes and wearing minimal clothes to the nearest 0.01 kg, and the height was measured to the nearest 0.1 cm with a stadiometer (Seca 769 with measuring rod 220 cm, Hamburg, Germany) [27]. The circumference of the waist, midarm, hips, and calf was measured according to standard procedures by using an anthropometric non-elastic tape (Lufkin W606PM; Apex Tool Group, Sparks, MD, USA) [28]. Also, the body mass index (BMI) was calculated [29].

#### 2.3.2. Body Composition

Patients came to the hospital refraining from any type of exercise, caffeine (tea, coffee, and energy drinks), and alcohol the day before the body composition measurements. In addition, they had fasted for at least 2–3 h (no foods, no liquids). Moreover, shoes, socks, tights, or everything else which could affect the measurement was removed, and females were in the middle period of menstruation [30]. Body composition measurements were conducted with the use of skinfolds and Bioelectrical Impedance (BIA). Specifically, skinfold thickness measurements were performed with a Slim Guide caliper (Creative Health Products, Ann Arbor, MI, USA) at a standing position according to standard methodology [27]. Skinfolds included three spots: triceps, subscapularis, and gastrocnemius, measuring to the nearest of 0.2 mm. Also, via BIA Body Fat (BF) (percentage and kg), Total Body Water (TBW), Extracellular Water (ECW) and Intracellular Water (ICW) (percentage and liters), Lean Body Mass (LBM), Dry Lean Body Mass (DLBM) (in kg), and Free Fat Mass Index (FFMI), Body Fat Mass Index (BFMI) were estimated. The measurement was performed via bioelectrical impedance Bodystat^®^ 1500MDD Bodystat 1500 (Bodystat Ltd., Douglas, Isle of Man, UK). The participants, during the measurement, stayed relaxed in the supine position, connecting the two electrodes on the hand and feet, respectively, as proposed by the manufacturer. The appendicular lean mass (ALM) was estimated from published equations by using weight, height, sex, and raw Resistance values measured from bioelectrical impedance analysis [31].

#### 2.3.3. Physical Performance Measurements

A physical performance assessment was conducted in a separate room of the hospital and included four tests. Before the measurements, patients completed a standardized eight minutes warm-up, which included two cycles of five full-body dynamic stretching exercises. The duration and the interval of each exercise were 30 and 15 s, respectively. Afterward, the patients started the following performance tests. 

30-s Chair Stand Test

To perform the 30-s chair stand test, the participants were seated on a chair with their arms crossed at the chest level and their hands over their shoulders. They had to stand up from this seated position until they reached a complete knee extension and then sit down again until the back touched the backrest of the chair. This procedure was repeated as many times as they could in 30 s [32].

One Leg Standing Test (OLST) for Both Legs

Participants were asked to stand on either their left or right leg and were instructed to keep their legs from touching and to maintain a single-leg stance for as long as possible, recording each try with a stopwatch. The try began once the foot was lifted off the floor and ended when the lifted foot was placed on the floor or with arm movements and the placement of their hand on the wall, which was in front of them for support if they needed it. Each leg was tested three times unless subjects performed perfectly on the first two trials [33].

6-m Walk Test (Slow and Fast)

Each participant completed two trials, a slow and a fast, starting at the standing position. Patients were instructed to stand with their toes touching the taped start line and walk at their usual (slow) and fast pace to a few steps beyond the taped finish line. Patient’s performance was recorded from the moment their foot crossed the start line to the moment their foot crossed the finish line [34].

Hand-grip Strength Test (HGS)

HGS was measured using a digital hand-grip strength meter (WCS-99.9, Beijing Xindong Huateng Sports Equipment Co., Ltd., Beijing, China) with a range of 5 to 99.9 kg. Before the test, participants adapted to the grip with some probationary tries. They performed three maximal intensity tries, with 45″ intervals between them, for each hand, calculating the average kilos. During the measurement, participants kept their bodies upright, feet naturally apart, arms diagonally down, palms facing inward, and maintaining an upright posture [35].

### 2.4. Malnutrition Assessment

Global Leadership Initiative on Malnutrition (GLIM) tool

The nutritional status was assessed with the GLIM tool [36]. In order to diagnose malnutrition, at least one phenotypic criterion along with one etiologic criterion should be fulfilled [36]. All subjects fulfilled the etiologic criteria due to disease burden/inflammation. For the purposes of the present study, the following three different phenotypic criteria were considered:(i)low BMI (i.e., BMI < 20 k kg/m^2^ if <70 yrs or BMI < 22 kg/m^2^ if ≥70 yrs, indicating moderate malnutrition; BMI < 18.5 kg/m^2^ if <70 yrs or BMI< 20 kg/m^2^ if ≥70 yrs, indicating severe malnutrition) [36].(ii)low CC.(iii)low FFMI < 17 kg/m^2^ for men and <15 kg/m^2^ for women [36].
Malnutrition Inflammation Risk Tool (MIRT)


The MIRT tool was developed by Jansen et al. in 2016 and was found to predict clinical outcomes in CD patients, including CD-related flares, hospitalizations, and surgeries [12]. It is a score from 0 to 8, with increased values denoting higher malnutrition and inflammation, and it was calculated as a sum of points gathered from (i) + (ii) + (iiii) [12]:(i)BMI: >20 kg/m^2^ (0 points), 28.5–20 kg/m^2^ (1 point), <18.5 kg/m^2^ (2 points).(ii)Weight loss in the previous 3 months: <5% (0 points), 5–10% (2 points), ≥10% (3 points).(iii)CRP: <5 (0 points), 5–50 (2 points), ≥50 (3 points).

### 2.5. Sarcopenia Evaluation

To evaluate sarcopenia, the European Working Group on Sarcopenia2 in Older People (EWGSOP2) criteria were used, namely hand strength (<27 kg and <16 kg, for men and women, respectively), peripheral lean body mass (appendicular lean mass, ALM) (<20 kg and <15 kg for men and women respectively) and gait speed (speed ≤ 0.8 m/s) [26].

### 2.6. Questionnaires

The participants, during their therapy with biological agents, answered the questionnaires, which were read and filled out by one volunteer, making it easier for them to give the answers. The following questionnaires were used: (i) the modified Harvey Bradshaw Index (MHBI) [37] to examine the level of the disease’s remission or exacerbation, (ii) the Sarcopenia Questionnaire (SARC-F) [38] for the examination of the incidence of sarcopenia, (iii) the Lawton Instrumental Activities of Daily Living Scale (LIADLS) [39] and (iv) the Activities Daily Living Scale (ADLS) [40] to record patients’ functional ability for complex and basic daily activities, respectively and (v) the Mini Nutritional Assessment (MNA) questionnaire (short form) [41].

### 2.7. Biomarkers and Other Variables Assessed

CRP (high sensitivity), albumin, and creatinine were measured. Data on the disease activity, duration, localization, and behavior were recorded. Moreover, data regarding the extended small intestine disease, peri-proctal disease, enterectomy, extra-intestinal manifestations, endoscopic response, and endoscopic healing were recorded (dichotomous variables, yes or no). Smoking habits were also recorded as a dichotomous variable (smoker, non-smoker). 

### 2.8. Statistical Analysis

The Kolmogorov-Smirnoff test was used to test the normality. The values are shown as means, standard deviation (SD) for normally distributed variables, or median with interquartile range (IQR) for non-normally distributed variables. The independent sample *t*-test (normally distributed variables or normally distributed variables after appropriate transformation) or Mann-Whitney test (non-normally distributed variables) was used for possible differences between the genders. Pearson’s or Spearman correlation coefficients were used, for normally distributed and skewed variables/patterns, respectively. Also, linear regression analysis was conducted to find out the percentage of prediction of independent to dependent variables.

A Principal Components Analysis (PCA) was applied for the identification of body composition/function patterns. More than one component with eigenvalues (originating from the correlation matrix of the standardized variables) was retained for the data analyses. The scree plot was also used to check the previous decision. The orthogonal varimax rotation was applied to generate optimal, non-correlated patterns. Five components of body composition/function patterns were identified based on the principle that the higher absolute scores indicate the variables contributing most to a particular component (absolute loading value > 0.6). Linear regression models were developed to evaluate the association of the extracted body composition/function components (independent variables) on log CRP (dependent variable) after adjustment for age. It is noted that CRP was transformed in a log form to achieve normality. Moreover, Spearman correlation coefficients were applied to test the association of the derived patterns with the MNA score. The analysis was performed with the statistical software IBM SPSS Statistics for Windows, Version 27.0. Armonk, NY, USA: IBM Corp. The level of significance was set at *a* = 0.05.

### 2.9. Sample Size Calculation

The prediction of the sample size (n) occurred with G*Power 3.1.9.7 (Universität Düsseldorf, Düsseldorf, Germany) for Windows [42]. Determining a large effect size at 0.51 for two groups (Males and Females) ascertain that a sample size of 39 participants, giving a 95% probability of rejecting the null hypothesis, will be needed to detect significant differences and correlations.

## 3. Results

The basic characteristics of the subjects are shown in Table 1. Fifty-three patients (26 women and 27 men) aged 38.1 ± 10.9 years with CD were studied. The age range was from 21 to 58 and 18 to 58 years old for men and women, respectively. The median duration of the disease was 9.0 years (25th–75th, 4.0–11.0). Seventy-nine-point one percent of patients were in remission, 14.6% had mild disease, and 6.3% had moderate disease severity with an MHBI score of 2 ± 2 (mean ± SD). Men had significantly higher weight, HGS and dry lean weight and lower gait speed, body fat mass and BFMI than women (*p* < 0.05). No differences were observed in OLST (R), and phase angle (Table 2). Females appeared to be less hydrated, having higher fat mass indexes than men (Table 2).

According to the participant’s medical history, they were following a single or combined therapy of aminosalicylates, corticosteroids, immunomodulators, and biological agents. The median (interquartile range) for the LIADLS and ADLS was 8 (8–8) and (6–6), respectively. Indeed, all patients achieved the maximum score on both scales, suggesting a good functional ability for complex and basic daily activities. It is noted that disability was defined as an exclusion criterion. Moreover, they were physically active at least two times per week with activities such as walking, running, swimming, football, volleyball, tennis, etc.

In Table 3, the malnutrition and sarcopenia prevalence in patients with CD is presented. As can be seen, the prevalence of malnutrition was different according to the used criteria of the GLIM algorithm ranging from 5.6 to 11.3%. The highest malnutrition prediction was observed with the use of FFMI in the GLIM criteria. It is noted that the prevalence of malnutrition was higher in women with the use of FFMI in the GLIM criteria. Most subjects scored low on the MRIT tool, with the vast majority having a score of 0 on a scale of 0–8. Three subjects had a score of 1, and one subject had a score of 2.

In Table 4, the factor loadings of PCA analysis are presented briefly; five patterns of body composition/function patterns were identified based on the principle that the higher ab-solute scores indicate the variables contributing most to a particular component (absolute loading value > 0.6). Pattern 1:MAC, FFMI, Max HGS; Pattern 2: Gait speed; Pattern 3: CC, age; Pattern 4: Chair standing test, OLST; Pattern 5: Albumin, TSF.

In Table 5, the linear regression analysis results are shown with logCRP as a dependent variable and the identified body composition/function patterns as independent ones. The pattern of increased albumin and TSF was inversely related to CRP levels after the adjustment for sex (R^2^ = 23.7%). It is noted that the model remained significant after further adjustment for disease severity. The linear regression model was also applied to a subsample of patients being in remission in order to avoid potential bias pertaining to disease severity. The pattern of increased albumin and TSF was significant logCRP “prediction” after adjustment for age (B = −0.279, SE = 0.095, R^2^ = 45.6%). Furthermore, a Spearman correlation was performed between the identified body composition/function patterns and MNA total score as well its constructive components (MNA sub-scores) (Table 6). As shown, Pattern 4 (Chair-standing test, OLST) was positively related to the MNA sub-score regarding food intake. Moreover, Pattern 1 (MAC, FFMI, Max HGS) and Pattern 3 (CC, age) were positively related to the MNA sub-score pertaining to BMI. When considering only patients in remission, small differentiations in the above associations were observed. More particularly, the MNA total score was related to Pattern 3 (CC, age) (rho = 0.417, *p* = 0.043), the MNA sub-score for food intake was related to Pattern 5 (Alb, TSF) (rho = 0.455, *p* = 0.025), the MNA sub-score of weight loss was associated to Pattern 1 (MAC, FFMI, Max HGS) (rho = −0.404, *p* = 0.050) and the MNA sub-score pertaining to BMI was also related to Pattern 1 (rho = 0.700, *p* < 0.001) (Appendix A).

## 4. Discussion

In the present study, the rate of malnutrition varied according to the GLIM criteria used, and its detection rate increased with the use of more accurate indicators, such as the FFMI. Sarcopenia was not detected in the present sample, while a pattern with high albumin and TSF identified with the use of PCA analysis was related to reduced CRP levels.

In general, patients with CD have a high risk of malnutrition due to loss of appetite, reduced food intake, impaired absorption, increased gastrointestinal losses, possibly increased energy requirements, and side effects of medication [8,9]. The early detection and management of malnutrition are of major importance since malnutrition is connected to postoperative complications [43], increased infection rates, and worse prognosis [10]. Several criteria for the detection of malnutrition in patients with IBD have been proposed, as recently reviewed [10]. The reported rates of malnutrition range from 12 to 85% [11], with patients in remission having lower rates of malnutrition [44] and high heterogeneity being evident between studies [45].

In the present study, the rate of malnutrition was relatively low compared to other studies [11]. This finding can be explained by the criteria used and/or the study population characteristics as well as the followed therapy. More particularly, few studies have assessed malnutrition with the GLIM criteria, so comparability between studies is relatively limited [13,14]. In addition, other tools have low sensitivity in malnutrition diagnosis when compared to the GLIM criteria, although similar results have been reported for GLIM and MIRT [46]. In parallel, the studied population appears to be at low nutritional risk. Indeed, as proposed by ESPEN, severe nutritional risk in patients with IBD is defined as the presence of at least one of the following: (i) weight loss > 10–15% within six months, (ii) BMI < 18.5 kg/m^2^ or (iii) albumin < 30 g/L [8]. In the present study, subjects had none of the aforementioned criteria, and the vast majority of patients were in a remission phase (42 out of 53 patients), while the rest had mild (8 out of 53 patients) or moderate symptoms (3 out of 53 patients). Possibly, by examining the nutritional characteristics of our sample, a balanced dietary plan could prevent malnutrition and, by extension, the presence of sarcopenia, especially in patients with CD in remission. 

It is noted that in other studies, including patients in remission, participants were also characterized as well-nourished [47]. Patients’ characteristics and disease history may also play a role in the observed low rate of malnutrition. For example, previous surgery, which predisposes patients to malnutrition [10], was performed in 17 out of 53 patients. The medical treatment of patients (in remission) is a major explanatory parameter of the good nutritional status of the patients and a reversal factor for the presence of sarcopenia. Indeed, a direct effect of anti-TNF treatment on body weight and body composition has been previously reported [48,49]. It is also possible that age plays a pivotal role in the development of sarcopenia. For example, in the study of Ünal et al., sarcopenia and probable sarcopenia were present in ~40% of patients with CD in remission [18]. The difference in the presence of sarcopenia between our study and that of Ünal et al. may be explained by the difference in the age range of participants (38.2 ± 10.9 vs. 49.4 ± 14.5 y). In a study including Greek participants of similar age (41.3 ± 14.1 y) to the present sample (38.2 ± 10.9 y), the prevalence of sarcopenia was 5% [50]. Indeed, as previously demonstrated, patients with possible sarcopenia were older compared to patients without sarcopenia (54.5 ± 15.9 vs. 47.4 ± 13.0 y) [18]. Nevertheless, a higher inflammatory burden increases sarcopenia risk even in young subjects (mean age 29.9 y) [51]. 

Women presented higher rates of malnutrition according to the GLIM criteria in the present sample. Gender differences have not adequately been addressed in the literature, but women with CD have been reported to have lower FFMI [52] and to be more frequently malnourished (lower protein, folate, vitamin A, thiamin, and calcium intakes) [53], while they may face additional nutritional challenges, in periods of pregnancy or lactation [54]. Women may score worse regarding subjective disease measurements [55]. In addition, hormonal perturbations during the menstrual cycle may worsen GI symptoms, such as diarrhea [54], while other diseases may more frequently co-exist, such as osteoporosis [56], possibly turning into osteosarcopenia [57]. In parallel, alterations in inflammatory indices may be differentiated by gender in healthy volunteers [58,59], although differences were not documented in the present work. Differences in drug responsiveness are also possible [60]. Moreover, the anti-TNF therapy used in patients with CD [24], as well as strength training [61], leads to more muscle gain in males than in females. Indeed, clinicians and dietitians should be more vigilant while treating women with CD in order to detect features of malnutrition and/or sarcopenia.

Regarding BMI, body composition values and physical performance measurements may be adversely affected in patients with CD, even in patients in remission in some [62] but not all studies [50]. BMI, the most frequently reported anthropometric variable, has been found lower in IBD patients than controls [63,64], while overweight and obesity may also be present in patients with IBD, paralleling the increased obesity rates in the general population [65] or implying an aetiopathogenic role of visceral fat in gut inflammation [66]. Nevertheless, in patients with increased BMI, the risk of complications decreases [65], and lower rates of penetrating disease behavior have been reported [67], confirming the obesity paradox. In the present study, the median BMI was normal (25.0 kg/m^2^), but with the BMI-based GLIM phenotypic criteria, three subjects still had problematic values. Compared to other studies, including IBD patients in remission, our results are comparable to a study reporting a mean BMI of 26.2 kg/m^2^ [68] or higher than a study conducted in Greek adults (mean BMI in patients in remission 17.1 kg/m^2^) [50] or other studies reporting a mean BMI of 22.1 kg/m^2^ [69]. Regarding FFMI, our results were comparable to those of other studies with CD patients (median value 18.7 kg/m^2^ in our study vs. 17.3 kg/m^2^) [52]. Women had lower values of FFMI in line with other studies in CD patients [52]. Fat mass was higher than that reported by others (20 kg vs. 14.4 kg) [69], which may be explained by the lower BMI values of patients in that study [69]. The same was evident for TSF [63].

The assessment of muscle functionality is crucial in both CD [70], and sarcopenia conditions since changes in muscle function can be detected earlier than structural changes [71]. Indeed, the HGS test has been found to predict the functional nutritional status in patients with CD [72,73]. In the present study, HGS was slightly lower or similar to that reported by others in patients in remission [72,74]. Regarding the other physical performance measurements, there is great variability in the methodology used (different tests, different time intervals, or protocols), rendering direct comparisons difficult [73,75,76,77]. Similarly, the same “problem” has been identified when considering sarcopenia assessment in this subgroup of patients [17]. Patients with CD have comparable or less physical activity than the general population [70,78,79]. Daily life activities have been reported to be low in CD patients. In the present study, the functional ability was very high in the participants, as assessed by two different scales [25,39]. However, these scales have been traditionally tested in older individuals [80]. Moreover, in the present population, the patients were in remission, and patients with physical difficulties were excluded from recruitment. It is noted that in studies including patients in remission, similar results have been documented according to daily life physical activities [81,82].

Regarding sarcopenia, which combines the phenotypes of muscle quantity and function in its definition, it is estimated that fifty percent of patients with CD have sarcopenia [16]. However, estimates may vary according to sarcopenia definition, age, and ethnicity, increasing variability in sarcopenia percentages ranging from 20 to 70% [17]. The SARC-F questionnaire has been designed to screen for patients at risk of sarcopenia (case-finding) but cannot identify early signs of the disease [62]. In a recent study, the probability of a false-negative assessment of sarcopenia was more than 60% [83]. In this sample, SARC-F scores were very low, and sarcopenia was undetectable. In another study, in Greek patients with IBD, sarcopenia was 2.2% in the total sample and 1.4% in patients in remission [50], which in part confirms our results. It is, however, noted that the assessment of sarcopenia in our study was based on measured functionality tests and predictive equations of ALM, which originated from samples with different characteristics [31].

CRP correlates with IBD activity, and it is related to postoperative complications, especially in patients with CRP > 10 mg/L. [84]. In the present study, only three patients had such values. Moreover, CRP has been previously related to sarcopenia in patients with CD [51]. Albumin reflects nutritional and inflammatory status in patients with CD [10]. It is noted that albumin levels lower than 33.6 g/L and 35 g/L have been associated with postoperative complications [84] and anastomotic leakage [85], correspondingly. Hopefully, in the present phenotypically “well nourished” sample, the median and lower end (25th percentile) values of albumin were higher than the aforementioned critical values, which is in line with the low malnutrition rates. Interestingly, a pattern with high albumin and TSF identified in the present study was inversely related to the measured inflammatory circulating molecules, i.e., CRP. This relation potentially reflects the good nutritional status of patients having high albumin levels and higher TSF. Moreover, the inflammation–muscle mass interplay should be considered. For example, platelet activating factor (PAF), interleukin-6 (IL-6), or tumor necrosis factor-*a* (TNF-α) can increase muscle breakdown and reduce protein synthesis, while in cachexia and sarcopenia, the inflammatory burden is increased, implying a bi-directional relationship [86,87,88,89,90]. Indeed, CRP and albumin have been previously reported as useful markers for CD differential diagnosis [91] and CD activity [92], and their ratio predicts mucosal healing in patients with CD [93]. Thus, the present results suggest that a good nutritional status, as reflected both by biochemical markers and TSF, relates to an amelioration of the inflammatory milieu. 

It is also noted that several body composition/function patterns were detected instead of fewer that would simply categorize patients into those with bad/good function. This may be explained by the sample size and the sample characteristics. It is, however, also possible that each test reflects different angles of nutritional status. This hypothesis is in line with the fact that only selected correlations were significant regarding the identified patterns and MNA-score and its components. 

The strengths of the present study include the assessment of a plethora of anthropometric, body composition, and function parameters related to malnutrition and sarcopenia. In parallel, certain biochemical indices were measured, which enabled the association of inflammatory fluctuations with body composition/function patterns. 

Limitations of the present study include the relatively small but adequate sample size, as suggested by the power analysis performed. Given that certain variables had zero variability, no other comparisons/correlations could be performed. The cross-sectional nature of the present work cannot imply causality in the observed associations. In addition, the data-driven by nature of identified patterns limits the repeatability and “translation” of the present results in a different clinical context. It is also possible that other inflammatory indices may have been affected by alterations in body composition, such as Lp-PLA_2_, which is implicated in PAF metabolism [94], is altered in patients with CD [95,96,97], and is differentiated in various body composition patterns [98]. Such data would be interesting to investigate in the future, given the alterations of PAF and Lp-PLA_2_ in cachexia [86,87] and their influence from diet [99,100]. In the present study, we did not include a control group. It is noted that other studies in patients with CD in remission did not use a control group [18]. Moreover, the use of a control group in our case would include healthy participants with a similar age (38.18 y). In this age group, it is un-probable to detect malnutrition or sarcopenia in the absence of an underlying disease.

The present study is the first to be conducted using a variety of sarcopenia detection tools to determine the presence of sarcopenia in a sample of patients with CD in remission without making comparisons to a control group. This is in contrast to many studies where the prevalence of sarcopenia among patients with CD exceeds 40%. Furthermore, when comparing genders, females had a higher percentage of malnutrition compared to males. However, this difference was not related to the presence of sarcopenia in either gender.

Future research should be directed toward exploring novel perspectives that could streamline the detection of both sarcopenia and CD. One promising avenue is the utilization of the Mendelian randomization method, which assesses genetic variations and investigates the causal impact of a modifiable factor on disease within observational studies [101,102,103]. By incorporating such methodologies, potential recommendations for the treatment and prevention of these conditions could be proposed.

## 5. Conclusions

In conclusion, based on the studied anthropometric, nutritional, and functional variables, CD patients were not diagnosed with sarcopenia in the present study. Body composition patterns were related to the inflammatory burden, underlying the interplay of inflammation and malnutrition, even in remission states. Further studies on older populations and during disease exacerbation are necessary to explore the potential link between CD, inflammation, and sarcopenia. 

## Figures and Tables

**Table 1 nutrients-15-03615-t001:** Basic characteristics of the subjects.

	Total (n = 53)	Men (n = 27)	Women (n = 26)	*p* Value
Age (yrs)	38.18 (10.92)	34.65 (8.39)	41.26 (12.06)	0.039
BMI (kg/m^2^)	25.00 (23.30–29.40)	26.10 (23.80–27.50)	24.40 (21.27–31.25)	0.395
Waist circumference (cm)	96.48 (16.87)	99.60 (12.14)	93.78 (19.99)	0.265
Hip circumference (cm)	107.00 (100.00–114.00)	108.50 (103.70–113.00)	105.00 (97.00–127.00)	0.394
Smoking (yes)	n = 4 (7.5%)	n = 1 (3.7%)	n = 3 (11.5%)	0.322
Disease duration (yrs)	9.00 (4.00–11.00)	10.00 (4.00–11.00)	7.50 (4.25–11.00)	0.396
Localization ∫				0.811
L1	n = 19 (35.80%)	n = 10 (37.0%)	n = 9 (34.6%)	
L2	n = 3 (5.7%)	n = 1 (3.7%)	n = 2 (7.7%)	
L3	n = 17 (32.1%)	n = 8 (29.6%)	n = 9 (34.6%)	
L4	n = 0 (0.0%)	n = 0 (0.0%)	n = 0 (0.0%)	
Behavior ∫				0.959
B1	n = 12 (22.6%)	n = 6 (22.2%)	n = 6 (23.1%)	
B2	n = 14 (26.4%)	n = 7 (25.9%)	n = 7 (26.9%)	
B3	n = 3 (5.7%)	n = 1 (3.7%)	n = 2 (7.7%)	
B2B3	n = 10 (18.9%)	n = 5 (18.5%)	n = 5 (19.2%)	
Extended small intestine disease (yes)	n = 15 (28.3%)	n = 7 (25.9%)	n = 8 (30.7%)	0.500
Periproctal disease (yes)	n = 11 (20.7%)	n = 4 (14.8%)	n = 7 (26.9%)	0.271
Enterectomy (yes)	n = 17 (32.0%)	n = 9 (33.3%)	n = 8 (30.7%)	0.444
Extraintestinal manifestations (yes)	n = 10 (18.8%)	n = 4 (14.8%)	n = 6 (23.0%)	0.394
Endoscopic response (yes) ∫	n = 35 (89.7%)	n = 17 (89.4%)	n = 18 (90.0%)	0.500
Endoscopic healing (yes) ∫	n = 24 (61.5%)	n = 11 (57.8%)	n = 13 (65.0%)	0.450
Albumin (g/dL) ‡	4.45 (4.28–4.65)	4.57 (4.37–4.77)	4.36 (4.19–4.50)	0.040
CRP (mg/dL) ‡	0.13 (0.07–0.32)	0.10 (0.04–0.22)	0.16 (0.08–0.41)	0.127
Creatinine (mg/dL)	0.67 (0.59–0.80)	0.62 (0.51–0.84)	0.74 (0.61–0.80)	0.478

BMI: Body mass index; CRP: C-reactive protein; Localisation: L1: ileum; L2 colonic; L3 ileocolonic; L4 upper gastrointestinal tract; Behavior: B1 non-structuring non-penetrating; B2 structuring; B3: penetrating; B2B3: both penetrating and structuring disease either at the same or different times; ∫ missing data for 14 patients, the percentages have been calculated from a total sample of 39 subjects; ‡ The variable was logarithmized prior to statistical comparison Data are presented as mean (standard deviation) for normally distributed variables and median and interquartile range (25th–75th) for non-normally distributed variables. Categorical variables are presented as frequencies (n, %). The independent sample *t*-test (normally distributed variables or normally distributed variables after appropriate transformation) or Mann-Whitney test (non-normally distributed variables) was used for possible differences between the genders.

**Table 2 nutrients-15-03615-t002:** Body composition and physical performance measurements of the volunteers.

	Total (n = 53)	Men (n = 27)	Women (n = 26)	*p* Value
Fat mass (kg)	20.40 (15.35–30.50)	17.60 (12.50–23.80)	24.95 (17.35–35.32)	0.022
Body fat (%)	28.66 (10.22)	21.03 (7.36)	35.25 (7.50)	0.001
Fat free mass (kg)	57.25 (12.60)	69.13 (5.32)	47.10 (6.40)	<0.001
FFMI (kg/m^2^)	18.73 (2.89)	20.87 (1.93)	16.88 (2.22)	<0.001
ALM * (kg)	19.30 (4.49)	23.49 (1.82)	15.67 (2.44)	<0.001
TBW (%)	40.51 (7.97)	54.40 (5.40)	47.40 (5.74)	0.001
ECW(lt)	18.22 (2.95)	21.50 (5.44)	16.71 (3.54)	0.002
ECW (%)	22.87 (2.57)	22.80 (2.74)	22.90 (2.84)	<0.001
ICW (%)	22.59 (5.44)	30.15 (5.21)	25.44 (3.85)	0.002
ICW (lt)	27.62 (4.93)	27.25 (2.44)	18.63 (3.83)	0.001
Phase angle (^o^) ‡	5.70 (5.15–6.65)	6.10 (5.40–7.10)	5.65 (4.67–6.32)	0.395
CC (cm)	35.20 (4.10)	35.29 (4.36)	35.13 (3.99)	0.907
CS (mm)	8.20 (5.01)	7.08 (4.99)	9.07 (4.97)	0.225
MAC (cm)	33.51 (3.93)	34.47 (3.10)	32.63 (4.44)	0.132
TSF (mm)	9.83 (5.98)	10.96 (5.56)	8.84 (6.26)	0.241
Chair standing test (replications)	15.52 (5.27)	15.80 (5.63)	15.27 (5.05)	0.751
Gait speed test (slow) (s for 6 m)	5.41 (1.13)	5.56 (1.17)	5.24 (1.10)	0.365
Gait speed test (high) (s for 6 m)	3.26 (2.88–4.28)	3.13 (2.70–4.12)	3.73 (3.17–4.68)	0.039
HGS—right (kg)	25.90 (20.15–39.05)	39.40 (35.62–43.05)	21.15 (18.77–25.62)	<0.001
HGS—left (kg)	25.00 (20.12–39.62)	39.65 (35.75–42.07)	20.60 (18.22–23.47)	<0.001
HGS—max (kg)	26.60 (21.22–40.97)	41.05 (38.55–45.07)	22.30 (19.20–25.62)	<0.001
OLST—right (s)	32.67 (19.73–55.02)	32.67 (24.55–66.98)	32.48 (8.78–51.40)	0.351
OLST—left (s)	31.63 (19.53–64.86)	33.50 (21.85–65.87)	30.16 (5.85–64.86)	0.489

‡ The variable was logarithmized prior to statistical comparison. * ALM was estimated from published equations [31]. FFMI: Fat free mass index; ALM: Appendicular lean mass; TBW: Total body water; ECW: Extracellular water; ICW: Intracellular water; CC: Calf circumference; CS: Calf skinfold; MAC: Midarm circumference; TSF: Triceps skinfold; HGS: Hand-grip strength; OLST: One leg standing test. Data are presented as mean (standard deviation) for normally distributed variables. Otherwise, data are presented as the median and interquartile range (25th–75th). The independent sample *t*-test (normally distributed variables or normally distributed variables after appropriate transformation) or Mann-Whitney test (non-normally distributed variables) was used for possible differences between the genders.

**Table 3 nutrients-15-03615-t003:** Malnutrition and sarcopenia prevalence in patients with CD.

Criterion Used	Total (n = 53)	Men (n = 27)	Women (n = 26)	*p*-Value
GLIM				
GLIM (BMI phenotypic criteria)	n = 3 (5.6%)	n = 1 (3.7%)	n = 2 (7.6%)	0.556
GLIM (CC phenotypic criteria)	n = 4 (7.5%)	n = 2 (7.4%)	n = 2 (7.6%)	0.593
GLIM (FFMI phenotypic criteria)	n = 6 (11.3%)	n = 0 (0%)	n = 6 (23%)	0.017
MRIT (scale 0–8)				0.519
Score 0	n = 49 (92.4%)	n = 26 (96.2%)	n = 23 (92.3%)	
Score 1	n = 3 (5.6%)	n = 1 (3.7%)	n = 2 (7.6%)	
Score 2	n = 1 (1.8%)	n = 0 (0%)	n = 1 (3.8%)	
Score >2	n = 0 (0%)	n = 0 (0%)	n = 0 (0%)	
MNA—score	13.0 (12.0–14.0)	13.0 (12.0–14.0)	13.0 (10.0–14.0)	0.732
MNA categories				0.351
MNA—malnutrition (score 0–7)	n = 1 (1.8%)	n = 0 (0%)	n = 1 (3.8%)	
MNA—at risk of malnutrition (score 8–11)	n = 6 (11.3%)	n = 2 (7.4%)	n = 4 (15.3%)	
MNA—subscales				
MNA Food intake	2 (2–2)	2 (2–2)	2 (2–2)	0.049
MNA weight loss	3 (2–3)	3 (2–3)	3 (3–3)	0.210
MNA mobility	2 (2–2)	2 (2–2)	2 (2–2)	1.000
MNA acute stress	2 (0–2)	1.5 (2–2)	2 (0–2)	0.671
MNA neurological problems	2 (2–2)	2 (2–2)	2 (2–2)	0.112
MNA BMI	3 (3–3)	3 (2–3)	3 (2–3)	0.329
Sarcopenia—related variables				
SARC-F	0 (0–1)	0 (0–1)	0 (0–1)	0.697
Low HGS * (yes)	n = 4 (7.5%)	n = 2 (7.4%)	n = 2 (7.6%)	0.659
Low ASM (yes) §	n = 8 (15%)	n = 0 (0%)	n = 8 (30.7%)	0.003
Low gait speed (slow) (yes) ∫	n = 3 (5.6%)	n = 1 (3.7%)	n = 2 (7.6%)	0.384
Low gait speed (slow) (yes) ∫	n = 0 (0%)	n = 0 (0%)	n = 0 (0%)	NA
ECOGSW2 (yes)	n = 0 (0%)	n = 0 (0%)	n = 0 (0%)	NA

Data are presented as median and interquartile range (25th–75th) or frequencies (n, %). GLIM: Global leadership initiative on malnutrition; BMI: Body mass index; CC: Calf Circumference; FFMI: Fat free mass index; MIRT: Malnutrition inflammation risk tool; MNA: Mini nutritional assessment; SARC-F: Sarcopenia questionnaire; HGS: Hand-grip strength; NA: Not applicable; * Low HGS: <27 kg for men and <16 kg for women; § Low ASM < 20 kg for men and <16 kg for women; ∫ cut-off speed ≤ 0.8 m/s.

**Table 4 nutrients-15-03615-t004:** Component loadings are derived using principal component analysis for the identification of body composition/function patterns.

	Components				
	Pattern 1 MAC, FFMI Max HGS	Pattern 2 Gait speed	Pattern 3 CC, Age	Pattern 4 Chair Standing Test, OLST	Pattern 5 Albumin, TSF
Albumin (log) (mg/dL)	0.216	0.205	−0.137	−0.191	0.787
CC (cm)	0.385	−0.164	0.757	0.172	−0.179
MAC (cm)	0.692	0.120	0.547	−0.106	0.160
TSF (mm)	−0.111	−0.056	0.096	0.316	0.697
Chair standing test (times/30 s)	0.065	−0.035	−0.107	0.829	−0.033
Gait Speed test (slow) (s)	0.111	0.958	0.084	0.013	−0.081
Gait Speed test (high) (s)	−0.139	0.923	0.064	−0.025	0.231
Max HGS (kg)	0.842	0.020	−0.248	0.012	−0.140
FFMI (kg/m^2^)	0.917	−0.099	0.197	0.021	0.163
OLST (right leg) (s)	−0.057	0.024	0.111	0.741	0.100
Age (years)	−0.202	0.291	0.785	−0.074	0.060

CC: Calf circumference; MAC: Mid-arm circumference; TSF: Triceps skinfold; HGS: hand-grip strength; FFMI: Fat free mass index; OLST: One-legged stance test.

**Table 5 nutrients-15-03615-t005:** Linear regression analysis with log CRP as the dependent variable and body composition/function patterns.

	Unstandardized Coefficients		Sig.
	B	Std. Error	
(Constant)	−0.759	0.144	0.000
Sex †	−0.080	0.268	0.769
Pattern 1 (MAC, FFMI, Max HGS)	−0.027	0.127	0.835
Pattern 2 (Gait speed)	0.004	0.083	0.957
Pattern 3 (CC, MAC, age)	−0.051	0.084	0.548
Pattern 4 (Chair-standing test, OLST)	0.074	0.080	0.363
Pattern 5 (Alb, TSF)	−0.180	0.085	0.046

† Males = 1, females = 0; MAC: Mid-arm circumference; FFMI: Fat free mass index; HGS: hand-grip strength; CC: Calf circumference; OLST: One-Legged Stance Test Triceps Skinfold; Alb: Albumin; TSF: Triceps skinfold.

**Table 6 nutrients-15-03615-t006:** Spearman correlation coefficients between MNA score, MNA sub-scores, and body composition/function patterns.

	Pattern 1 (MAC, FFMI, Max HGS)	Pattern 2 (Gait Speed)	Pattern 3 (CC, MAC, Age)	Pattern 4 (Chair-Standing Test, OLST)	Pattern 5 (Alb, TSF)
MNA total score	0.177	0.142	0.301	−0.006	0.009
	*p*= 0.351	*p* = 0.455	*p* = 0.106	*p* = 0.974	*p* = 0.964
MNA Food intake	0.292	0.111	0.182	0.377	0.209
	*p* = 0.110	*p* = 0.553	*p* = 0.328	*p* = 0.037	*p* = 0.260
MNA weight loss	−0.326	0.123	−0.141	−0.116	0.171
	*p* = 0.074	*p* = 0.509	*p* = 0.450	*p* = 0.533	*p* = 0.358
MNA mobility	NA	NA	NA	NA	NA
MNA acute stress	0.116	0.208	0.278	0.116	0.077
	*p* = 0.535	*p* = 0.261	*p* = 0.130	*p* = 0.535	*p* = 0.680
MNA neurological problems	0.253	0.015	0.156	0.078	−0.220
	*p* = 0.170	*p* = 0.936	*p* = 0.401	*p* = 0.678	*p* = 0.235
MNA BMI	0.544	−0.094	0.382	−0.061	−0.034
	*p* = 0.002	*p* = 0.616	*p* = 0.034	*p* = 0.744	*p* = 0.854

MAC: Mid-arm circumference; FFMI: Fat free mass index; HGS: hand-grip strength; CC: Calf circumference; OLST: One-Legged Stance Test; Alb: Albumin; TSF: Triceps Skinfold; MNA: Mini Nutritional Assessment; BMI: Body Mass Index. NA: Not applicable. It is noted that no variability was detected for MNA mobility, and thus no correlation coefficient could be calculated.

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
