# Peer review of "Nutritional Risk and Sarcopenia Features in Patients with Crohn’s Disease: Relation to Body Composition, Physical Performance, Nutritional Questionnaires and Biomarkers"

_nutrients, 2023, doi:10.3390/nu15163615_

Round 1
Reviewer 1 Report
The manuscript entitled „Nutritional Risk and Sarcopenia Features in Patients with Crohn's Disease: Relation to Body Composition, Physical Performance, Nutritional Questionnaires and Biomarkers” presents interesting issue, but some problems should be corrected.
Major:
1. Authors applied controversial procedure, as they combined in one studied group patients in remission and exacerbations. While they all have Crohn's Disease, their clinical status, symptoms and observed conditions are totally different, as well as the diet which they follow and physical activity (during the study) are totally different. Especially weight loss in the previous 3 months is supposed to differ between patients in remission and exacerbations, so those groups should not be combined. Taking this into account, Authors should rather study only the remission patients (being the majority of their studied group) as the homogenic population, which would be a valuable contribution.
2. Authors indicate that they diagnosed sarcopenia, but they did not diagnosed it based on proper diagnostic criteria. Authors should get familiar with the current recommendations within clinical algorithm used for sarcopenia (https://www.ncbi.nlm.nih.gov/pmc/articles/PMC6322506/), which indicates that 3 elements should be assessed to diagnose sarcopenia: muscle strength, muscle quantity and physical performance, as without 3 elements combined we are unable to diagnose sarcopenia properly. Authors did not assess 3 required elements, but only 2 of them (hand strength and peripheral lean body mass), so they can not state that they diagnosed sarcopenia – in fact they are unable to diagnose sarcopenia based on this element only.
Abstract:
Authors should formulate more specific conclusions – for the Crohn's Disease, as the formulated conclusion “selected criteria differentiated in malnutrition detection” is not associated with the studied group and aim of the study.
Introduction:
Authors should present the information in more organized way. In the present version of the manuscript they include some random information, e.g. associated with diet, which do not justify the conducted study (lines 50-53)
Authors should focus on presenting the current state of knowledge and the gaps in existing knowledge, while they should include the results of the already conducted studies on IBD patients (CD and UC ones) to indicate what is already known, what is not known, and why their study may bring any novel information.
Materials and methods:
There is a problem with the studied group (in one studied group combined patients in remission and exacerbations) – as described above. Authors should focus on remission individuals
“confirmed CD patients” – how was CD confirmed?
The detailed method of bioelectrical impedance measurement should be presented (e.g. metal elements, menstrual cycle, physical activity on the proceeding day, etc.) – according to the recommendations of the measurement
There is a problem with assessment of sarcopenia (see above).
Results:
Instead of focusing on the comparison between women and men, Authors should focus on any CD-specific comparison, e.g. depending on the location of disease, behavior, or disease duration.
Discussion:
The discussion should focus more on the implications of the observed results – for the therapy and dietary recommendations.
Conclusions:
Authors should formulate more specific conclusions – for the Crohn's Disease, as the formulated conclusion “selected criteria differentiated in malnutrition detection” is not associated with the studied group and aim of the study.
Author Contributions:
It seems that contribution of some of Authors (PN, CG, CPI) was only minor and they did not participate in preparing manuscript. There is a serious risk of a guest authorship procedure which is forbidden. In such case (if they did not participate in manuscript preparation) they should be rather presented in Acknowledgements Section and not be indicated as authors of the study.
Authors should properly define the contributions – e.g. what do Authors mean by “validation” if they did not conduct any validation in their study? There are similar doubts in case of “resources” and “data curation”
Author Response
Reviewer 1
Dear reviewers
We really appreciate the spend time for the judge of our manuscript. Your indications helped us to improve it.
C: Comment
A: Answer
- Authors applied controversial procedure, as they combined in one studied group patients in remission and exacerbations. While they all have Crohn's Disease, their clinical status, symptoms and observed conditions are totally different, as well as the diet which they follow and physical activity (during the study) are totally different. Especially weight loss in the previous 3 months is supposed to differ between patients in remission and exacerbations, so those groups should not be combined. Taking this into account, Authors should rather study only the remission patients (being the majority of their studied group) as the homogenic population, which would be a valuable contribution.
- Thank you for your important comment. We understand that disease activity is an important factor to consider in the present study. Indeed, the vast majority of patients were in remission as indicated from 1) the modified Harvey Bradshaw index, 2) therapy, 3) low level of malnutrition, and 4) daily life physical activities.
More specifically:
- Lines 262 – 264: “Seventy-nine-point one percent of patients were in remission, 14.6% had mild disease and 6.3% had moderate disease severity with MHBI score 2±2 (mean ± SD).” Therefore, according to the modified Harvey Bradshaw index and sample’s clinical condition, none patient had severe disease and/or exacerbation, thus, they had controlled symptoms. Moreover, there was not any difference between the sample’s measurements, thus, we continued checking the possible differences according to gender.
- Lines 290 – 291: “The sample was following a single or combined therapy of aminosalicylates, corticosteroids, immunomodulators and biological agents.” Therefore, the sample was under control regarding the clinical status.
- Line 374: “The low level of malnutrition”, Lines: 383 – 388: In parallel, the studied population appears to be in low nutritional risk. Indeed, as proposed by ESPEN, severe nutritional risk in patients with IBD is defined as the presence of at least one of the following: i) weight loss >10–15% within six months, ii) BMI <18.5 kg/m2 or (iii) albumin <30 g/L [8]. In the present study subjects had none of the aforementioned criteria and the vast majority of patients were in a remission phase.”
- Lines 455 – 458: “Moreover, in the present population most the patients were in remission and patients with physical difficulties were excluded in recruitment. It is noted that in studies including patients in remission similar results have been documented according daily life physical activities [80,81].” Thus, the sample had similar physical activity level.
We have also updated the manuscript to better illustrate patients’ characteristics.
Specifically, we rephrased the “2.1 participants” adding more specific details about the inclusion and exclusion criteria of our sample. Lines 94 – 102: “Participants (n=53, 27 men and 26 women) were recruited from a public Greek hospital. Inclusion criteria were (a) confirmed CD patients in remission or mild or moderate symptoms, (b) aged ≥ 18 years old and (c) capable to self–serve in their daily routine activities, (d) physical active at least two times per week, (e) controlled diet and (f) in pharmaceutical treatment [22]. Exclusion criteria were (a) CD patients in exacerbation, (b) age less than 18 years old, (c) co-presence of other autoimmune disease, (d) difficulties in daily living. All measurements and questionnaire filling were done in Theageneio anti-cancer hospital, from May to August of 2022.”
Also, we extended the discussion clarifying the sample’s clinical status. Lines 386 – 391: “In the present study subjects had none of the aforementioned criteria and the vast majority of patients were in a remission phase (42 out of 53 patients), while the rest had mild (8 out of 53 patients) or moderate symptoms (3 out of 53 patients). Possibly, by examining the nutritional characteristics of our sample, a balanced dietary plan could prevent malnutrition and, by extension, the presence of sarcopenia, especially in patients with CD in remission.”
Furthermore, in order to test our findings exclusively in patients with remission we ran the presented linear regression models only in this subsample (n=42 patients in remission). The results remained the same, and their significance was more pronounced (R2=45.6%). This observation was also added at the results section.
|
Coefficientsa |
||||||
|
Model |
Unstandardized Coefficients |
Standardized Coefficients |
t |
Sig. |
||
|
B |
Std. Error |
Beta |
||||
|
1 |
(Constant) |
-.648 |
.180 |
|
-3.597 |
.002 |
|
MAC, HGS, FFMI |
.028 |
.145 |
.059 |
.192 |
.850 |
|
|
Gait speed |
.049 |
.098 |
.096 |
.494 |
.628 |
|
|
CC, MACM age |
-.013 |
.083 |
-.029 |
-.151 |
.882 |
|
|
chairstand, OLST |
.101 |
.096 |
.200 |
1.052 |
.308 |
|
|
Alb, TSF |
-.279 |
.095 |
-.582 |
-2.923 |
.009 |
|
|
Sex_0=females, 1=males |
-.210 |
.303 |
-.227 |
-.693 |
.498 |
|
|
a. Dependent Variable: log_CRP |
||||||
When considering only patients in remission, small differentiations in the above associations were observed. More particularly, the MNA total score was related to Pattern 3 (CC, age) (rho=0.417, p=0.043), the MNA sub-score for food intake was related to Pattern 5 (Alb, TSF) (rho=0.455, p=0.025), the MNA sub-score of weight loss was associated to Pattern 1 (MAC, FFMI, Max HGS) (rho= -0.404, p=0.050) and the MNA sub-score pertaining to BMI was also related to Pattern 1 (rho=0.700, p<0.001) (Supplementary Table 1).
|
|
Pattern 1 (MAC, FFMI, Max HGS) |
Pattern 2
(Gait speed) |
Pattern 3
(CC, MAC, age) |
Pattern 4 (Chair-standing test, OLST) |
Pattern 5
(Alb, TSF) |
|
MNA TOTAL SCORE |
0.215 |
0.110 |
0.417 |
0.066 |
0.178 |
|
p=0.312 |
p=0.608 |
p=0.043 |
p=0.761 |
p=0.405 |
|
|
MNA Food intake
|
0.306 |
0.092 |
0.332 |
0.366 |
0.455 |
|
p=0.147 |
p=0.670 |
p=0.113 |
p=0.078 |
p=0.025 |
|
|
MNA weight loss
|
-0.404 |
0.291 |
-0.048 |
-0.162 |
0.210 |
|
p=0.050 |
p=0.168 |
p=0.822 |
p=0.451 |
p=0.325 |
|
|
MNA mobility
|
NA |
NA |
NA |
NA |
NA |
|
MNA acute stress |
0.044 |
0.106 |
0.367 |
0.180 |
0.193 |
|
p=0.840 |
p=0.623 |
p=0.078 |
p=0.399 |
p=0.367 |
|
|
MNA neurological problems |
0.256 |
-0.045 |
0.346 |
-0.166 |
-0.166 |
|
p=0.227 |
p=0.834 |
p=0.097 |
p=0.439 |
p=0.439 |
|
|
MNA BMI |
0.700 |
-0.112 |
0.347 |
-0.045 |
0.116 |
|
P<0.001 |
p=0.602 |
p=0.097 |
p=0.836 |
p=0.588 |
- Authors indicate that they diagnosed sarcopenia, but they did not diagnosed it based on proper diagnostic criteria. Authors should get familiar with the current recommendations within clinical algorithm used for sarcopenia (https://www.ncbi.nlm.nih.gov/pmc/articles/PMC6322506/), which indicates that 3 elements should be assessed to diagnose sarcopenia: muscle strength, muscle quantity and physical performance, as without 3 elements combined we are unable to diagnose sarcopenia properly. Authors did not assess 3 required elements, but only 2 of them (hand strength and peripheral lean body mass), so they cannot state that they diagnosed sarcopenia – in fact they are unable to diagnose sarcopenia based on this element only.
- We would like to thank the reviewer for this comment. Indeed, we have used the cutoffs of the EWGSOP2 criteria (Low HGS:<27 kg for men; <16 kg for women and low appendicular lean mass (< 15 kg for men and <20 kg for women) as the reviewer suggests. We have now provided data on low gait speed and related cut-offs as suggested by EWGSOP2 criteria (<=0.8 m/sec).
The prevalence of sarcopenia by using all three criteria i.e., strength, peripheral lean body mass and gait speed was zero (no individual had all of these criteria). For consistency reasons, the manuscript has been updated to appropriately reflect all three EWGSOP2 criteria.
We have also used additional tests reflecting physical performance, such as the 30 – second chair stand test and the one- leg standing test (OLST), for both legs. Several of the aforementioned measurements can be used individually or in combination for the detection of sarcopenia, particularly in longitudinal screening. The number of diagnostic criteria is wide, and we hold great respect for each method. However, the conclusions drawn by Cruz-Jentoft et al. (2019) and the study by Ackermans et al. (2022) indicate that there is poor agreement between definitions. Despite the wide variety of tools available, clinical decision-making is not straightforward. Taking into consideration the aforementioned literature, your comment, and the methodology we followed, we have highlighted the importance of several sarcopenia-related measurements.
Lines 110 – 115: “Considering the extensive array of screening tools [18,23] and measurements for detecting sarcopenia, in alignment with the new trends set by the European Working Group on Sarcopenia in Older People (EWGSOP2) [23,26] and the absence of a confirming structured approach for determining which tool to utilize, Mseveral measurements were taken regarding anthropometric, body composition physical performance tests, malnutrition assessment, nutritional questionnaires, biomarkers and other variables.”
Ackermans, L. et al Clinical nutrition ESPEN, 48, 36–44. https://doi.org/10.1016/j.clnesp.2022.01.027.
Cruz-Jentoft, A. J., et al. Age and ageing, 48(1), 16–31. https://doi.org/10.1093/ageing/afy169.
- Authors should formulate more specific conclusions – for the Crohn's Disease, as the formulated conclusion “selected criteria differentiated in malnutrition detection” is not associated with the studied group and aim of the study.
- We modified the abstracts’ conclusion. Lines 36 – 41: “In conclusion, based on the studied anthropometric, nutritional and functional variables, CD patients were not diagnosed with sarcopenia in the present study. Body composition patterns were related to inflammatory burden, underlying the interplay of inflammation and malnutrition, even in remission states. Further studies on older populations and during disease exacerbation are necessary to explore the potential link between CD, inflammation, and sarcopenia.”
C . Authors should present the information in more organized way. In the present version of the manuscript they include some random information, e.g. associated with diet, which do not justify the conducted study (lines 50-53)
Authors should focus on presenting the current state of knowledge and the gaps in existing knowledge, while they should include the results of the already conducted studies on IBD patients (CD and UC ones) to indicate what is already known, what is not known, and why their study may bring any novel information.
A . We modified the introduction adding some extra references which are similar with our study. Moreover, we have deleted some nutritional studies, which are not directly relevant to our results.
Lines 64 – 85: “Malnutrition can result in loss of muscle mass and function and possibly sarcopenia [15], defined by changes in muscle quantity and functionality. Indeed, it is estimated that ~ 50 % of patients with CD have sarcopenia [16], while estimates may vary according to sarcopenia definition, age and ethnicity from 20 to 70% [17]. It isAs documented by Ünal et al., a considerable proportion of IBD patients (142 out of 344) in clinical remission, who are malnourished or at risk of malnutrition have a high rate of sarcopenia and probable sarcopenia [18]. Similarly, Ryan et al. showed that 42% of the studied IBD patients had sarcopenia [19]. Moreover, sarcopenia as a direct result of chronic inflammation and malnutrition, has both diagnostic and prognostic significance in IBD patients [20], it hinders patient’s ability for postoperative recovery, it increases the likelihood of surgical complications [19], and is generally associated to adverse patient outcomes [21]. In addition, the associated muscle wasting and weakness results in fatigue and reduced quality of life, both of which are prevalent in people living with CD [4,22].
However, there are severely bias and flaws in that kind of studies. A main flaw is the heterogeneity of the sarcopenia assessment, because of the plethora of existing detection tools [23]. Additionally, a noteworthy factor pertains to patients with CD, who are administered anti-TNF agents like infliximab [24]. These individuals exhibit a reversal of symptoms associated with inflammatory sarcopenia and muscle wasting [24]. Hence, it is essential to conduct concurrent screening for nutritional status and body composition analysis in patients with IBD. This approach ensures the provision of suitable nutritional support, even during the remission period, and helps prevent sarcopenia-related surgical complications and unfavorable clinical outcomes [18].”
C . There is a problem with the studied group (in one studied group combined patients in remission and exacerbations) – as described above. Authors should focus on remission individuals “confirmed CD patients” – how was CD confirmed?
A . Thank you for your comment. We have thoroughly addressed your prior comment. Moreover, the confirmation of remission in patients with CD was established through the use of the Harvey Bradshaw index, with scores of less than 4 points signifying a state of remission.
Lines 262 – 264: Seventy-nine-point one percent of patients were in remission, 14.6% had mild disease and 6.3% had moderate disease severity with MHBI score 2±2 (mean ± SD).
Furthermore, as described above, the results did not show differentiation even when the analysis was performed only on patients with Crohn's disease in remission. Additionally, Crohn's disease was confirmed by the gastroenterologist who examined patients' clinical images, symptoms, and the depiction of the colon after colonoscopy.
C . The detailed method of bioelectrical impedance measurement should be presented (e.g. metal elements, menstrual cycle, physical activity on the proceeding day, etc.) – according to the recommendations of the measurement.
A . Thank you for the comment. We included these details to ensure the quality of our measurement. Lines 121 – 128: “Patients came to the hospital refraining from any type of exercise, and caffeine (tea, coffee and energy drinks) and alcohol the day before the body composition measurements. alcohol or stimulant beverages. In addition, Tthey had fasted for at least 2 – 3 hours (no foods, no liquids). Moreover, shoes, socks, tights or everything else which could affect the measurement was removed and females were in the middle period of menstruation [30].”
C . Instead of focusing on the comparison between women and men, Authors should focus on any CD-specific comparison, e.g. depending on the location of disease, behavior, or disease duration.
A . Thank you for your suggestion. Nevertheless, we have opted to solely focus on gender comparisons due to the constraints posed by the number of subgroups you recommended. As a reminder of the earlier response, "no disparities were observed in the measurements of the sample, prompting us to further investigate potential variations based on gender. An elucidation of this potential limitation is provided in the discussion section. Lines 503 – 505: “Limitations of the present study include the relatively small but adequate sample size, as suggested by the performed power analysis. Given that certain variables had zero variability, no other comparisons/correlations could be performed”
In addition, the differentiations between genders detected in the present study is of additive value to the literature and clinical practice. Indeed, clinicians and dietitians should be more vigilant while treating women with CD in order to detect features of malnutrition and/or sarcopenia.
C . The discussion should focus more on the implications of the observed results – for the therapy and dietary recommendations.
A . We included therapy and dietary recommendations.
Lines 386 – 408. “In the present study subjects had none of the aforementioned criteria and the vast majority of patients were in a remission phase (42 out of 53 patients), while the rest had mild (8 out of 53 patients) or moderate symptoms (3 out of 53 patients). Possibly, by examining the nutritional characteristics of our sample, a balanced dietary plan could prevent malnutrition and, by extension, the presence of sarcopenia, especially in patients with CD in remission.
. It is noted that in other studies including patients in remission, participants were also characterized as well-nourished [46]. Patients’ characteristics and disease history may also play a role in the observed low rate of malnutrition. For example, previous surgery, which predisposes patients in malnutrition [10], was performed in 17 out of 53 patients. The medical treatment of patients (in remission) is a major explanatory parameter of the good nutritional status of the patients and a reversal factor for the presence of sarcopenia. Indeed, a direct effect of anti-TNF treatment on body weight and body composition has been previously reported [47,48]. It is also possible that age plays a pivotal role in the development of sarcopenia. For example, in the study of Ünal et al. sarcopenia and probable sarcopenia were present in ~ 40% of patients with CD in remission [18]. The difference in the presence of sarcopenia between our study and that of Ünal et al may be explained by the difference in age range of participants (38.128 ± 10.92 vs 49.4 ± 14.5 y). In a study including Greek participants of similar age (41.3 ± 14.1 y) to the present sample (38.218 ± 10.92 y) the prevalence of sarcopenia was 5% [49]. Indeed, as previously demonstrated patients with possible sarcopenia were older compared to patients without sarcopenia (54.5 ± 15.9 vs 47.4 ± 13.0 y) [18]. Nevertheless, a higher inflammatory burden increases sarcopenia risk even in young subjects (mean age 29.9 y) [50].”
We have further highlighted the importance of sex differences in clinical practice. Lines 409 – 423: “wWomen presented higher rates of malnutrition according to theto the GLIM criteria in the present sample. Gender differences have not adequately been addressed in the literature, but women with CD have been reported to have lower FFMI [51], to be more frequently malnourished (lower protein, folate, vitamin A, thiamin, and calcium intakes) [52], while they may face additional nutritional challenges, in periods of pregnancy or lactation [53]. Women may score worse regarding subjective disease measurements [54]. In addition, hormonal perturbations during menstrual cycle may worsen GI symptoms, such as diarrhea [53], while other diseases may more frequently co-exist, such as osteoporosis [55] possibly turning into osteosarcopenia [56]. In parallel, alterations in inflammatory indices may be differentiated by gender in healthy volunteers [57,58], although differences were not documented in the present work. Differences in drug responsiveness are also possible [59]. Moreover, the anti-TNF therapy used in patients with CD [24] as well as strength training [60] leads to more muscle gain in males than females. Indeed, clinicians and dietitians should be more vigilant while treating women with CD in order to detect features of malnutrition and/or sarcopenia.”.
C . Authors should formulate more specific conclusions – for the Crohn's Disease, as the formulated conclusion “selected criteria differentiated in malnutrition detection” is not associated with the studied group and aim of the study.
A . We modified the study’s conclusion. Lines 534 – 539: “In conclusion, based on the studied anthropometric, nutritional and functional variables, CD patients were not diagnosed with sarcopenia in the present study. Body composition patterns were related to inflammatory burden, underlying the interplay of inflammation and malnutrition, even in remission states. Further studies on older populations and during disease exacerbation are necessary to explore the potential link between CD, inflammation, and sarcopenia”.
C . It seems that contribution of some of Authors (PN, CG, CPI) was only minor and they did not participate in preparing manuscript. There is a serious risk of a guest authorship procedure which is forbidden. In such case (if they did not participate in manuscript preparation) they should be rather presented in Acknowledgements Section and not be indicated as authors of the study. Authors should properly define the contributions – e.g. what do Authors mean by “validation” if they did not conduct any validation in their study? There are similar doubts in case of “resources” and “data curation”
A . It is evident that the authors mentioned should make contributions to the study. The three individuals you mentioned were involved in both the measurement process and the review process. We have included their names in the "Writing, Review, and Editing" section too.

Reviewer 2 Report
The authors aimed to investigate nutritional risk (malnutrition) and sarcopenia features in patients with Crohn's Disease.
I have some comments.
Study design: Please justify the absence of a control group (same age) in your study.
Methods: What was the range of age in men and women?
Results: As I understand, FFMI, ALM, handgrip strength, and gait speed were the main variables in the study. It would be interesting to see if there were any correlations between these parameters and disease-related traits, such as quality of life (for example: https://pubmed.ncbi.nlm.nih.gov/32168964/), the use of corticosteroids (with expected catabolic effects), disease duration, disease severity (remission / mild / moderate) etc.
Discussion: Please indicate what was the novelty of the study.
Discussion: Please explain why there were no patients with sarcopenia (for example, age is important, and we may expect high sarcopenia rates in 60+ people, but not in patients with age 30-45).
Discussion (perspectives): I recommend mentioning the future use of a Mendelian randomization method (for example: https://pubmed.ncbi.nlm.nih.gov/36904201/) to determine the impact of Crohn's disease on sarcopenia. For this, the bioinformaticians need to use genetic instruments related to CD (https://pubmed.ncbi.nlm.nih.gov/37156999/) and sarcopenia (https://pubmed.ncbi.nlm.nih.gov/36771461/).
Author Response
Reviewer 2
Dear reviewers
We really appreciate the time spent for the judge of our manuscript. Your indications helped us to improve it.
C: Comment
A: Answer
C . Please justify the absence of a control group (same age) in your study.
A . We included the absence of control group. Line 95. “…and were assessed without comparisons with a control group.” It is noted that other studies in patients with CD in remission did not use a control group (Ünal et al. 2021). Moreover, the use of a control group in our case would include healthy participants with a similar age (38.18 y). In this age group it is un-probable to detect malnutrition or sarcopenia in the absence of an underlying disease. This was also added in the limitations.
C . What was the range of age in men and women?
A . Lines 260 – 261. The results are presented as means and standard deviation. The age range was from 21 to 58 and 18 to 58 years old for men and women, respectively and has been now clarified in the text.
C . As I understand, FFMI, ALM, handgrip strength, and gait speed were the main variables in the study. It would be interesting to see if there were any correlations between these parameters and disease-related traits, such as quality of life (for example: https://pubmed.ncbi.nlm.nih.gov/32168964/), the use of corticosteroids (with expected catabolic effects), disease duration, disease severity (remission / mild / moderate) etc.
A . Thank you for the comment. The studied variables were CRP (high sensitivity), albumin and creatinine, the modified Harvey Bradshaw Index (MHBI), the Sarcopenia Questionnaire (SARC – F), the Lawton Instrumental Activities of Daily Living Scale (LIADLS), the Activities Daily Living Scale and the Mini Nutritional Assessment (MNA) questionnaire (short form), MIRT tool, Global Leadership Initiative on Malnutrition (GLIM) tool, 30 – second chair stand test, One leg standing test (OLST), for both legs, 6 – Meter walk test (Slow and Fast), Handgrip Strength test, skinfolds and bioelectrical impedance. As we mentioned in discussion (Lines 503 – 506: “Limitations of the present study include the relatively small but adequate sample size, as suggested by the performed power analysis. Given that certain variables had zero variability, no other comparisons or /and correlations could be performed. The cross-sectional nature of the present work cannot imply causality in the observed associations.”) it is a limitation of our study the difficulty to conduct more statistical analysis.
C . Please indicate what was the novelty of the study.
- Thank you for your helpful comment. We show the novelty of our study in many parts of the discussion (eg. lines 377, 410, etc.). However, we provide a resume of the novelty of our study in Lines 518 – 524: “The present study is the first to be conducted using a variety of sarcopenia detection tools to determine the presence of sarcopenia in a sample of patients with CD in remission, without making comparisons to a control group. This is in contrast to many studies where the prevalence of sarcopenia among patients with CD exceeds 40%. Furthermore, when comparing genders, females had a higher percentage of malnutrition compared to males. However, this difference was not related to the presence of sarcopenia in either gender.”
C . Please explain why there were no patients with sarcopenia (for example, age is important, and we may expect high sarcopenia rates in 60+ people, but not in patients with age 30-45).
A . We agree with your point. Indeed, the participants’ maximal age was 58 years old. A special reference to the role of age in developing sarcopenia in CD patients was also made (lines: 401 – 405). Moreover, independently the participants age, the biological agents, perhaps, provide a reversal effect in malnutrition and by extend in sarcopenia (lines: 388 – 391).
C . Discussion (perspectives): I recommend mentioning the future use of a Mendelian randomization method (for example: https://pubmed.ncbi.nlm.nih.gov/36904201/) to determine the impact of Crohn's disease on sarcopenia. For this, the bioinformaticians need to use genetic instruments related to CD (https://pubmed.ncbi.nlm.nih.gov/37156999/) and sarcopenia (https://pubmed.ncbi.nlm.nih.gov/36771461/).
- Thank you for the suggestion. We included some aspects of this new method of CD and sarcopenia detection at the end of the “discussion”. Lines 535 – 530: “Future research should be directed towards exploring novel perspectives that could streamline the detection of both sarcopenia and CD. One promising avenue is the utilization of the Mendelian randomization method, which assesses genetic variations and investigates the causal impact of a modifiable factor on disease within observational studies [100–102]. By incorporating such methodologies, potential recommendations for the treatment and prevention of these conditions could be proposed”

Round 2
Reviewer 2 Report
The revised version of the paper has been improved.